# EVADE CHATGPT DETECTORS VIA A SINGLE SPACE

## ABSTRACT

ChatGPT brings revolutionary social value but also raises concerns about the misuse of AI-generated text. Consequently, an important question is how to detect whether texts are generated by ChatGPT or by human. Existing detectors are built upon the assumption that there are distributional gaps between human-generated and AI-generated text. These gaps are typically identified using statistical information or classifiers. Our research challenges the distributional gap assumption in detectors. We find that detectors do not effectively discriminate the semantic and stylistic gaps between human-generated and AI-generated text. Instead, the "subtle differences", such as *an extra space*, become crucial for detection. Based on this discovery, we propose the SpaceInfi strategy to evade detection. Experiments demonstrate the effectiveness of this strategy across multiple benchmarks and detectors. We also provide a theoretical explanation for why SpaceInfi is successful in evading perplexity-based detection. And we empirically show that a phenomenon called *token mutation* causes the evasion for language model-based detectors. Our findings offer new insights and challenges for understanding and constructing more applicable ChatGPT detectors.

## 1 INTRODUCTION

In May 2023, news broke that attorney Steven A. Schwartz, with over 30 years of experience, employed six cases generated by ChatGPT in a lawsuit against an airline company. Remarkably, when requested about their accuracy, ChatGPT claimed they were entirely true. However, the judge later discovered that all six cases contained bogus quotes and internal citations, resulting in Schwartz being fined 5000 dollars. This alarming incident exemplifies the misuse of AI-generated text.

The advent of large language models like ChatGPT has undeniably created substantial social value (Felten et al., 2023; Zhai, 2022; Sallam, 2023b). Yet, alongside the positive impact, cases like Schwartz's highlight pressing concerns. AI-generated text has been found to be incorrect, offensive, biased, or even containing private information (Chen et al., 2023; Ji et al., 2023; Li et al., 2023; Lin et al., 2022; Lukas et al., 2023; Perez et al., 2022; Zhuo et al., 2023; Santurkar et al., 2023). Concerns regarding the misuse of ChatGPT span across various domains, such as education (Kasneci et al., 2023), healthcare (Sallam, 2023a), academia (Lund & Wang, 2023), and even the training of large-scale language models themselves.

A 2019 report by OpenAI (Solaiman et al., 2019) revealed that humans struggle to distinguish AI-generated text from human-written text and are prone to trusting AI-generated text. Consequently, relying on automated detection methods is an important effort in differentiating between human-generated and AI-generated text (Jawahar et al., 2020), spurring researchers to invest significant effort into this issue.

These detection methods typically assume the existence of *distributional gaps* between human-generated and AI-generated text, with detection achieved by identifying these gaps. We divide the detection methods into white-box and black-box detection. *White-box* detection methods leverage or estimate the intrinsic states of the text to model the distributional gaps, incorporating word distributions (Gehrmann et al., 2019), probability curvatures (Mitchell et al., 2023), and intrinsic dimensions (Tulchinskii et al., 2023). However, recent AI models like ChatGPT are often black boxes with inaccessible and hard-to-estimate internals, rendering white-box detection inapplicable. Thus, *black-box detectors* have also been proposed. These detectors primarily learn the distributional gap

by training binary classifiers with manually collected human corpora and AI-generated text (Guo et al., 2023; Solaiman et al., 2019; Tian et al., 2023).

Our work challenges the traditional understanding of the distributional gaps. We discover that detectors do not primarily rely on these gaps, at least not on those visible to humans in terms of semantics and styles. First, we find that even when generated text includes the phrase "As an AI model", detectors may still classify it as human-generated. This suggests that detectors do not properly utilize semantic information for detection. Second, we find that general style transfer is ineffective in evading detectors; only when the new style is highly intense can detection potentially be evaded.

Our experiments reveal that detectors rely on subtle text differences, such as an extra space. To demonstrate this, we propose a simple evasion strategy: *adding a single space character before a random comma* in the AI-generated text. Surprisingly, our method significantly reduces the detection rate for both white-box and black-box detectors. For GPTZero (while-box) and HelloSimpleAI (black-box) detectors, the proportion of detected AI-generated text drops from roughly 60%-80% to nearly 0%. The results are depicted in Fig. 2.

We endeavor to elucidate the efficacy of the strategy. We found the strategy induces a phenomenon termed as *token mutation*. This phenomenon results in the disappearance of a prevalent token, such as a comma, from the tokenized ids, transmuting it into a low-frequency token. The fundamental reason for this occurrence is the discrepancies in representations, implying that subtle alterations in text perceptible to humans can be significantly divergent for language models. From this observation, we extend and propose a series of infiltration methodologies, verifying the impacts of different alterations.

We summarize our major contributions as follows:

- We investigate how existing ChatGPT detectors leverage the distributional differences between AI-generated and human-generated text for detection. We find that detectors do not effectively exploit semantics and style-based content distributions.
- We propose an evasion strategy that involves inserting a single space character. This strategy performs well across multiple benchmarks and various types of detectors, revealing that ChatGPT detectors rely more on subtle formal discrepancies for detection.
- We explain why the space insertion strategy is effective. In particular, we provide a phenomenon called "token mutation" that causes the infiltration.

## 2 RELATED WORK

In this section, we discuss prior work related to the detection of AI-generated text, adversarial learning, and style transfer.

**Detection of AI-generated Text** The detection of AI-generated text has garnered significant attention in recent years (Jawahar et al., 2020), with various methods proposed to distinguish between human-generated and AI-generated text. White-box detection methods focus on the internal states of language models, using features such as word distributions(Gehrmann et al., 2019), probability curvatures(Mitchell et al., 2023), and intrinsic dimensions (Tulchinskii et al., 2023). However, these methods are not applicable to black-box language models like ChatGPT. Black-box detection methods, which have also been explored, train binary classifiers using human corpora and AI-generated text (Guo et al., 2023; Solaiman et al., 2019; Tian et al., 2023). Our work challenges the assumptions of these methods, showing that traditional distributional gaps may not be the primary factors used by detectors.

Previous work has already identified initial concerns regarding the robustness of ChatGPT detectors (Wang et al., 2023). However, the study has been limited to cross-domain and cross-lingual generalization capabilities. In contrast, this paper is the first to provide a comprehensive approach to evading detection by these detectors.

**Adversarial Learning** Adversarial learning has been widely studied in the field of computer vision, where classifiers can be fooled by making minor modifications to input images (Akhtar & Mian, 2018; Goodfellow et al., 2014; Szegedy et al., 2014). These modifications, known as adversarial perturbations, are often imperceptible to humans but can lead to misclassifications by the model.

Our work draws parallels between adversarial learning in computer vision and our findings in the context of AI-generated text detection. We demonstrate that a simple modification, such as adding a space character, can effectively evade detectors.

Adversarial learning has also been explored in the context of natural language processing (Ebrahimi et al., 2018; Jia & Liang, 2017). These studies often involve crafting adversarial examples for text classifiers, with the aim of improving model robustness or exposing vulnerabilities.

**Style Transfer** Style transfer techniques have been employed to transform text content while preserving its semantic meaning (Fu et al., 2018; Shen et al., 2017; Li et al., 2018). In our study, we examine the effectiveness of style transfer in evading AI-generated text detectors. We find that general style transfer is ineffective for this purpose, and only when the new style is highly intense can detection potentially be avoided. This result highlights the limitations of relying solely on style transfer to evade detection of AI-generated text.

## 3 SPACE INFILTRATION

We propose a method of space character attack to bypass AI text detectors. Specifically, we propose to add a space character before a random comma in the text. For example, in Fig. 1, given the user question "Describe the structure of an atom.", we first use ChatGPT to generate a response. Such response is likely to be detected as AI-generated. Then, with our SpaceInfi strategy, we add a new space before a random comma. If the response contains multiple paragraphs, we apply this strategy to each paragraph. In this case, the "charge," becomes "charge␣,", which results in a high probability to be detected as human-generated.

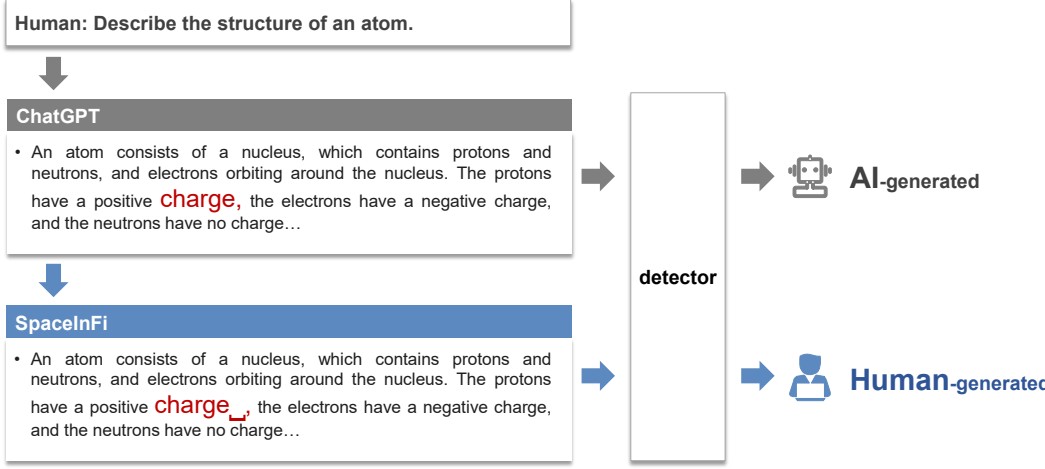

Figure 1: To evade detectors, SpaceInfi adds a space character before a randomly selected comma in the AI-generated text. (␣ indicates a space character.)

In addition to its simplicity, this approach has the following characteristics: (1) free, requiring no additional cost; (2) no loss of quality and imperceptibility. The new text has the same generated quality as the original text. Since the modification only involves adding a single space, it is unlikely to be noticed by a human. (3) The attack is model-agnostic, requiring no knowledge of the internal states of the LLMs or detector. In this paper, we denote this strategy as SpaceInfi (Space Infiltration).

## 4 EXPERIMENTS

### 4.1 BASELINES

In addition to employing SpaceInfi, we also considered several baselines.

**Act like a human** First, we are interested in whether ChatGPT knows how to evade detection on its own. We explicitly instruct ChatGPT to respond like a human and attempt to avoid being detected by the detector. Specifically, the prompt we use is as follows:

```
Question:  $QUESTION$

Requirement:  Answer the question like a human and avoid being
found that the answer was generated by chatGPT.
```

**Style transfer** ChatGPT has the ability to switch different answer styles based on different roles, such that their distributions are also different. As detectors leverage the distribution difference between AI-generated and human-generated texts, we leverage response styles to synthesize different distributions. We investigated whether evading the detector is possible by switching styles. Specifically, we consider three different styles to transfer, ordered by their intensity as follows: colloquial style, slang style, and Shakespearean style. We aim to control the degree of the distribution gap through the intensity of the style and thus verify the correlation between the distribution gap and detector performance. The complete instructions are as follows:

- **Colloquial style**:

```
Question:  $QUESTION$
Requirement:  Using more colloquial expressions in the
response.
```

- **Slang style**:

```
Question:  $QUESTION$
Requirement:  Answer the question in slang style.
```

- **Shakespearean style**:

```
Question:  $QUESTION$
Requirement:  Answer the question in Shakespearean style.
```

## 4.2 SETUP

**Benchmarks:** We use the AI-generated text from the following benchmarks.

- **Alpaca** (Taori et al., 2023) is an instruction dataset generated based on ChatGPT and self-instruction (Wang et al., 2022). It initially comprises 175 seed instructions and is expanded to 52k instructions using ChatGPT. During the expansion process, Alpaca aims to ensure diversity in the set of questions. For our experiments, we randomly selected 100 Alpaca instructions as the test set.
- **Vicuna-eval** is the test set used by Vicuna (Chiang et al., 2023). It consists of 80 questions covering nine categories, such as writing, roleplay, math, coding, and knowledge. These questions are more diverse than Alpaca. We use this benchmark to validate the effect of SpaceInfi on more diverse questions and responses.
- **WizardLM-eval** is the test set used by WizardLM (Xu et al., 2023). This test set consists of 218 challenging questions and covers a diverse list of user-oriented instructions including difficult coding generation & debugging, math, reasoning, complex formats, academic writing, and extensive disciplines.
- **Alpaca-GPT4** (Peng et al., 2023) is a GPT-4 version of Alpaca, which is considered to has higher quality. We use this benchmark to validate the effect of SpaceInfi on GPT-4 generated text.

**Metric:** We utilize ChatGPT (turbo-3.5) to generate responses for Alpaca, Vicuna-eval, and WizardLM-eval datasets. To obtain AI-generated text, we employ various evasion detection strategies. For Alpaca-GPT4, we directly use the released GPT-4 responses (Peng et al., 2023) and then apply the SpaceInfi strategy. Subsequently, we ask each detector to classify the text as either AI-generated or human-generated. To assess the performance of the evasion strategies, we compute the ratio of text identified as human-generated. We denote this ratio as the **evasion rate** of the evasion strategy.

**Detectors:** We examined several ChatGPT detectors that had been publicly disclosed (Guo et al., 2023; Mitchell et al., 2023; Tian, 2022; Tian et al., 2023). As of June 16, 2023, the publicly available detectors among them include GPTZero, HelloSImpleAI (Guo et al., 2023), and MPU (Tian et al., 2023).

- **GPTZero** (Tian, 2022) is a white-box detector that relies on statistical information within the text. Its detection is based on the text's perplexity and burstiness scores. Perplexity measures the level of randomness in a text, while burstiness quantifies the variation in perplexity. We utilized its version tailored for ChatGPT.
- **HelloSimpleAI** (Guo et al., 2023) is a black-box detector based on a classifier. It utilizes AI-generated text and human-generated text to train a text classifier using RoBERTa. We utilized its English version.
- **MPU** (Tian et al., 2023) is a detector based on a multiscale positive-unlabeled (MPU) framework for AI-generated text detection. It rephrases text classification as a positive unlabeled (PU) problem. We utilized its English version.

### 4.3 RESULTS

We show the results on different benchmarks and detectors in Fig. 2.

**SpaceInfi is effective and generalizes across all benchmarks and both ChatGPT-3.5 and GPT-4.** On four benchmarks, SpaceInfi demonstrates outstanding evasion performance. For GPTZero and HelloSimpleAI, the evasion rate of the original *no-prompt* strategy is about 20%. With SpaceInfi, the rate increases to close to 100%. This verifies that adding one space character as in SpaceInfi is able to evade ChatGPT detectors. We also observed that SpaceInfi fails to evade detection by MPU.

**ChatGPT itself is unaware of how to evade detection.** We observed that the *act-like-a-human* strategy does not increase the proportion of text being identified as human-generated. On the contrary, for the majority of cases, the text is even more likely to be recognize as AI-generated. This suggests that ChatGPT does not inherently possess the ability to evade detection. This observation aligns with our expectations, as the training corpus of ChatGPT does not contain information on evading detectors. In Sec 4.4, we will further investigate the behavioral patterns of ChatGPT and the performance of detectors for the act-like-a-human strategy.

**Evading detection through style requires a intense style switching**. As shown in Fig. 2, a relatively mild colloquial style does not clearly increase the evasion rate in most cases. We need to employ more intense slang or Shakespearean styles to effectively evade detection. Clearly, the stronger the style, the more difficult it is for humans to accept the text. We believe that slang and Shakespearean styles are almost unacceptable in real-life situations. Therefore, evading detection through style switching is not a viable method in practical applications.

**Compared to creating distributional differences through style transfer, it is more effective to evade detection by generating subtle differences with SpaceInfi.** Clearly, the SpaceInfi strategy, which only inserts a single space character, has virtually no impact on the text of the response. On the other hand, even the slightest colloquial expression in style can have a noticeable effect on the text (e.g., word distribution). Therefore, generating distributional differences by SpaceInfi is much more effective than by style transfer. We will provide a more detailed case analysis in Table 1.

### 4.4 HOW DO EVASIONS AFFECT THE GENERATED TEXT?

In Table 1, we provide concrete examples to demonstrate the texts generated with different strategies. The texts reveal some interesting behaviors of ChatGPT detectors.

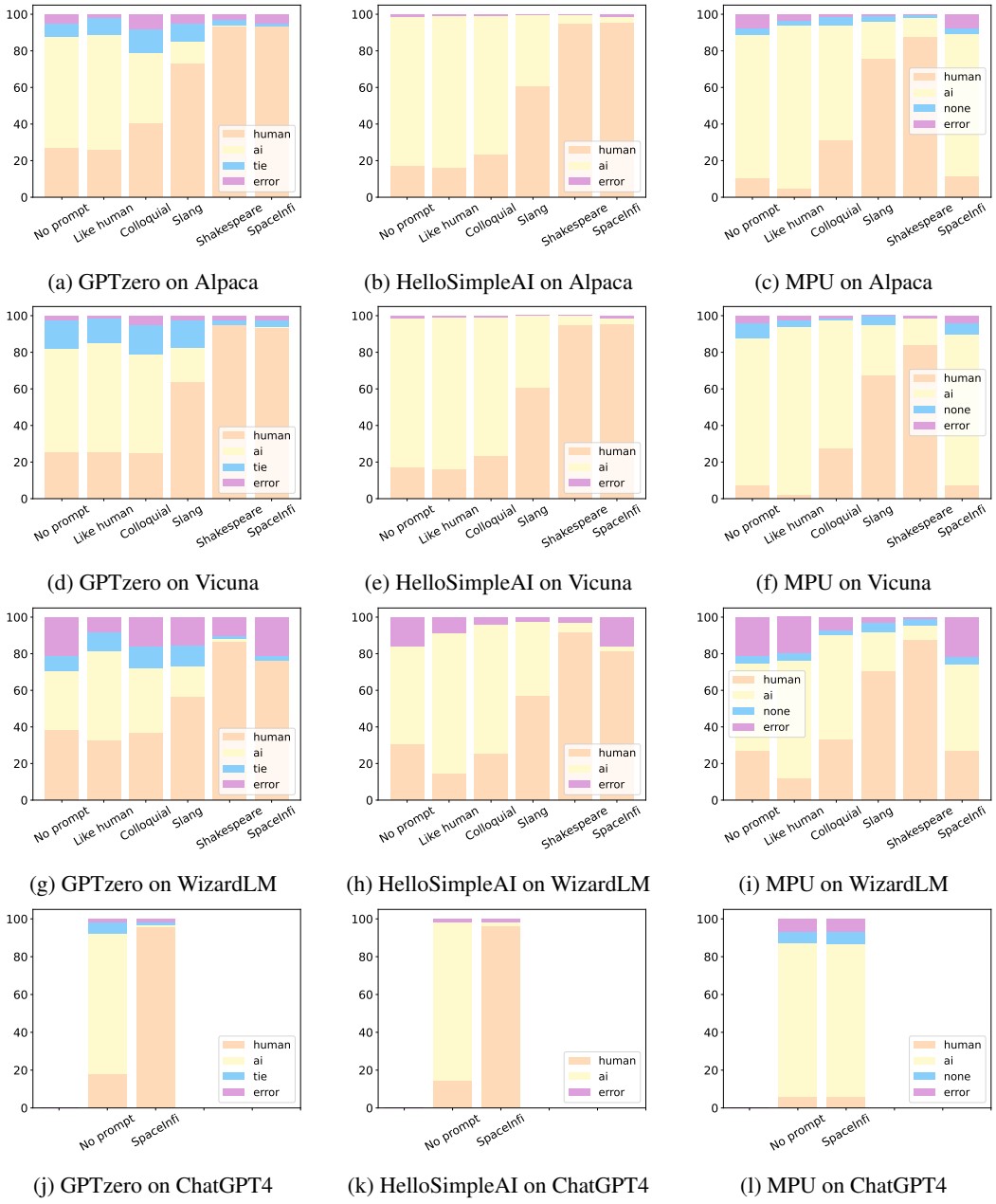

Figure 2: Results over different benchmarks and detectors. The detectors are tested on June 16, 2023.

Table 1: Texts Generated by Different Strategies.

| **Question:** | Describe the color pink. |
|---|---|

**No prompt** GPTZero: AI. HelloSimpleAI: AI. MPU: AI.

As an AI language model, I don't have personal experiences or visual perception. However, I can describe the color pink based on its widely recognized characteristics. Pink is a vibrant and soft hue that falls between red and white on the color spectrum. It is often associated with femininity, tenderness, and innocence. Pink can vary in intensity, ranging from pale pastel tones to deep and vivid shades. This color is frequently linked to affection, joy, and romanticism. It is commonly used in fashion, interior design, and branding to create a sense of charm and playfulness.

**Act like a human** GPTZero: Human. HelloSimpleAI: AI. MPU: AI.

Pink is a soft and gentle color, the lighter version of red. It's delicate and feminine, often associated with flowers, romance, and sweetness. It's a bright and cheerful color that can evoke feelings of warmth and happiness. It can be described as playful, charming, and cute. Depending on the shade, it can also be modern and bold. Overall, pink is a versatile and lovely color that can bring joy to anything it touches.

**Colloquial style** GPTZero: AI. HelloSimpleAI: AI. MPU: AI.

Pink is that color you see when you're walking through a garden. It's soft and delicate like the petals on a rose. It's also the color you see on a warm summer sunset, when the sky is a mix of pink and orange. Pink is the color of love, friendship and sweetness. It's light hearted and playful, like the color of cotton candy. A playful color that makes you smile, whether you're wearing it or staring at it. It's the perfect color for a sunny day picnic, where the strawberry pie is as sweet as the pink tablecloth it's on. That's what pink is – a soft, sweet color that makes everything a little brighter.

**Slang style** GPTZero: Tie. HelloSimpleAI: AI. MPU: AI.

Yo, pink be poppin' like a boss! It be that soft, cute, and flirty color that make you wanna blush. It's like a mix of red and white, with just the right amount of sweetness. Pink be the color that makes you feel girly and sassy at the same time. It's the color of unicorns, princesses, and all things fabulous. So, if you want to add some sass to your life, you got to add some pink!

**Shakespearean style** GPTZero: Human. HelloSimpleAI: Human. MPU: Human.

Oh! thou dost asketh of the hue of pink,
A shade so delicate, as roses think,
A tinge of red, with white's restraint,
On sight, it doth a blush maintain.

Its rosy glow, like dawn's first light,
Or twilight's ray, on summer's night,
A colour soft, sweet and mild,
With beauty, that leaves no heart beguiled.

It speaks of love, and tenderness much,
An essence, that's tender to the touch,
A color, so lovely to the eye,
Of shades, that doth one's spirit inspire high.

Ah! pink, a colour of grace divine,
A freshness, that doth ever shine,
An apt reflection of affection pure,
That's found, forevermore to endure.

**SpaceInfi** GPTZero: Human. HelloSimpleAI: Human. MPU: AI.

As an AI language model , I don't have personal experiences or visual perception. However, I can describe the color pink based on its widely recognized characteristics. Pink is a vibrant and soft hue that falls between red and white on the color spectrum. It is often associated with femininity, tenderness, and innocence. Pink can vary in intensity, ranging from pale pastel tones to deep and vivid shades. This color is frequently linked to affection, joy, and romanticism. It is commonly used in fashion, interior design, and branding to create a sense of charm and playfulness.

Firstly, the detector fails to leverage explicit semantic information for their detection. In this case, the responses of both no-prompt and SpaceInfi contain "As an AI language model". Interestingly, the SpaceInfi strategy still successfully evades the detector. This verifies that the detector is not sensitive to the semantics of the text. Therefore, **detectors do not rely on the semantic gap to differentiate human-generated and AI-genereated texts**.

Secondly, different strategies have different impact on the quality of the response. Evidently, the SpaceInfi strategy, which adds a single space, does not affect the original response quality. We also did not find clear impact of the act-like-a-human strategy on response quality. However, the style switch strategies do affect the response quality. That is, although the answers remain correct, their presentation becomes less acceptable. As the style intensifies, the acceptability of the answer format declines. According to the texts, SpaceInfi is the only strategy that retains the response quality and evasion rate.

## 5 WHY SPACEINFI WORKS?

Previous studies have suggested that the effect of detectors' classifications is due to the recognized differences between the AI-generated and human-generated texts. From this perspective, it is interesting to find that a extremely small distributional gap (i.e. a single space) by SpaceInfi successfully evade the detectors. In this section, we explain this phenomenon for different types of detectors, including while-box detectors (i.e. GPTZero) and black-box detectors (i.e. HelloSimpleAI).

### 5.1 WHY SPACEINFI IS EFFECTIVE FOR LM-BASED DETECTORS?

HelloSimpleAI uses the RoBERTa model (Liu et al., 2019) as the classifier backbone. RoBERTa is widely recognized for its strong generalization ability. Thus, it seems counter-intuitive that adding a single space could alter the classification outcome.

We have conducted a detailed investigation of the representations by RoBERTa for the texts before and after modification. As illustrated in Figure 3, we found that the tokens undergo mutations after modification. Typically, the token id for a comma "," is 6, while it is 2156 for a "␣,". The original comma token 6 has disappeared in the infiltrated text. Despite the high frequency occurrence of the ordinary comma id (6) in the corpus, the space comma (2156) is quite exceptional, especially within AI-generated texts.

| **text:** Hello**,** world! | **text:** Hello ␣**,** world! |
|:---:|:---:|
| **tokens:** 0, 31414, **6**, 232, 328, 2 | **tokens:** 0, 31414, **2156**, 232, 328, 2 |
| (a) Tokenizer ids for the original text. | (b) Tokenizer ids for the infiltrated text. |

Figure 3: Token Mutation Phenomenon. The two texts appear similar to humans, but for a LM-based detector, the token for "," has disappeared after modification, which makes its representation much different from the original text.

This implies that even though the differences between the two text segments may appear minimal to humans, there are substantial alterations in the language model representations due to the changes in token ids. The original token id has disappeared after modification. We refer to this phenomenon as *token mutation*. This fundamentally arises due to the mismatch in human understanding of the text and the language model's representation of the text based on tokenizers. Given the perennial nature of this mismatch, the attacks induced by token mutation have generality against detectors.

To justify the generality of token mutation, we present some of the token mutations we discovered in RoBERTa in Table 2. It is evident that although the difference between the two tokens may appear minimal to humans, there is a substantial alteration in token ids within the language models that the original token id has disappeared.

Subsequently, we selected three such token mutations and tested their capability to evade detectors when employed as attack mechanisms over Vicuna-eval and HelloSimpleAI. The results are demonstrated in Table 3. It can be observed that, similar to the original SpaceInfi strategy, these

Table 2: Examples of Token Mutation

| Token Mutation | Token Mutation |
|---|---|
| '.' (4) → '␣.' (479) | ',' (6) → '␣,' (2156) |
| '-' (12) → '␣-' (111) | ':' (35) → '␣:' (4832) |
| ')' (43) → '␣)' (4839) | '/' (73) → '␣/' (1589) |
| '"' (108) → '␣"' (128) | '""' (113) → '␣""' (22) |
| '?' (116) → '␣?' (17487) | ';' (131) → '␣;' (25606) |
| '%' (207) → '␣%' (7606) | '!' (328) → '␣!' (27785) |

token mutations invariably lead to a notable decline in detector capabilities. This corroborates the generality of the attacks inflicted by token mutation on the detectors.

Table 3: Comparison of Accuracy between Original Data and Token Mutation

| Strategy | Original accuracy | Accuracy after modification |
|---|---|---|
| ':' (35) → '␣:' (4832) | 81.3% | 9.4% |
| '"' (108) → '␣"' (128) | 81.0% | 33.4% |
| '␣' → '␣␣'(1437) | 80.8% | 9.6% |

We believe that the mismatched representations between humans and LMs causes the infiltration. As a result, similar minimal modifications may easily bypass LM-based detectors.

## 5.2 WHY SPACEINFI IS EFFECTIVE FOR PERPLEXITY-BASED DETECTORS?

We explain the reason from the mathematical formulation of perplexity. Perplexity is a measure of how well a probability language model predicts a natural language sentence. The perplexity of a sentence is computed by:

$$\text{Perplexity}(W) = \prod_{i=1}^{N} 2^{-\frac{1}{N} \log_2 p(w_i|w_{i-1},...,w_1)} \tag{1}$$

Here, $W$ represents the sentence, which consists of words $w_1, w_2, \ldots, w_N$, and $N$ is the number of words in the sentence. $p(w_i|w_{i-1}, \ldots, w_1)$ denotes the conditional probability of word $w_i$ given its past $i - 1$ words.

As SpaceInfi introduce an extra space, the perplexity contains a term

$$2^{-\frac{1}{N} \log_2 p(w_i=","|w_{i-1}=" ␣",...,w_1)} \tag{2}$$

We assume that AI-generated text is always well-formed. Specifically, when calculating perplexity, it did not encounter cases with extraneous spaces inserted. Therefore, $p(w_i = ","|w_{i-1} = " ␣", \ldots, w_1) \to 0$. It ultimately results in a high value for $Perplexity(W)$, leading the detector to consider the text as non-AI-generated.

This explains why SpaceInfi works for perplexity-based GPTZero. It also reveals why the robustness of the perplexity-based detector is low: we can easily modify the AI-generated text to obtain a very high perplexity.

## 6 CONCLUSION

In this paper, we have examined the efficacy of ChatGPT detectors and the implications of AI-generated text misuse. Our findings challenge the traditional understanding of the distributional gaps between human-generated and AI-generated text, revealing that detectors may not primarily rely on semantic and stylistic differences. We have demonstrated a simple evasion strategy by adding an extra space character before a random comma in AI-generated text, which significantly reduces the detection rate. We verify its effectiveness on a variety of benchmarks and detectors. We also explain its effect by revealing the token mutation phenomenon. Our observations underscore the challenges faced in developing robust and deployable ChatGPT detectors for open environments.

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
