# OpenReview forum: "Evade ChatGPT Detectors via A Single Space"
_ICLR.cc/2024/Conference — ICLR 2024 Conference Withdrawn Submission_

### Official Review · Reviewer_Gnd1 · 2023-10-26

**Soundness:** 3 good
**Presentation:** 3 good
**Contribution:** 3 good
**Rating:** 6
**Confidence:** 3

**Summary:**

This paper proposes a new strategy called SpaceInfi to evade detection by ChatGPT detectors using a single space. The authors challenge the assumption that there are distributional gaps between human-generated and AI-generated text and show that detectors do not primarily rely on these gaps. They find that even when generated text includes the phrase "As an AI model," detectors may still classify it as human-generated, suggesting that detectors do not properly utilize semantic information for detection. They also find that general style transfer is ineffective in evading detectors, only when the new style is highly intense can detection potentially be evaded. The SpaceInfi strategy has been shown to be effective in multiple benchmarks and detectors, offering new insights and challenges for understanding and constructing more applicable ChatGPT detectors.

**Strengths:**

This paper makes several significant contributions to the field of natural language processing and AI-generated text detection.

Firstly, the authors challenge the distributional gap assumption in detectors and show that detectors do not effectively discriminate the semantic and stylistic gaps between human-generated and AI-generated text. This finding is original and significant because it challenges a widely held assumption in the field and suggests that current detectors may not be as effective as previously thought.

Secondly, the authors propose the SpaceInfi strategy to evade detection, which is a novel and effective approach that relies on a single space to evade detection. This strategy is original and significant because it offers a new and effective way to evade detection that was not previously known.

Thirdly, the authors provide a theoretical explanation for why SpaceInfi is successful in evading perplexity-based detection and empirically show that a phenomenon called token mutation causes the evasion for language model-based detectors. This contribution is significant because it provides a deeper understanding of the mechanisms behind detection and evasion.

Fourthly, the authors investigate whether evading the detector is possible by switching styles and show that general style transfer is ineffective in evading detectors, only when the new style is highly intense can detection potentially be evaded. This finding is significant because it suggests that style transfer may not be an effective way to evade detection and highlights the importance of understanding the intensity of the style in relation to detector performance.

Overall, this paper is of high quality and clarity, with well-designed experiments and clear explanations of the findings. The contributions are significant and original, challenging existing assumptions and offering new insights and challenges for understanding and constructing more applicable ChatGPT detectors.

**Weaknesses:**

While this paper makes several significant contributions to the field, there are also some weaknesses that could be addressed to improve the work:

1. Limited scope of experiments: The experiments in this paper are limited to a few benchmarks and detectors. While the results are promising, it would be beneficial to expand the scope of the experiments to include more benchmarks and detectors to ensure the generalizability of the findings.

2. Lack of comparison with other evasion strategies: The paper only compares the SpaceInfi strategy with style transfer, but there are other evasion strategies that could be compared to SpaceInfi. For example, adversarial attacks or other methods that introduce subtle changes to the text could be compared to SpaceInfi to determine which strategy is most effective.

3. Lack of analysis of the impact of SpaceInfi on downstream tasks: While the paper shows that SpaceInfi is effective in evading detection, it does not analyze the impact of SpaceInfi on downstream tasks such as question answering or summarization. It would be beneficial to analyze the impact of SpaceInfi on these tasks to determine if the strategy has any unintended consequences.

4. Limited discussion of ethical implications: The paper briefly mentions the potential misuse of AI-generated text but does not delve into the ethical implications of the findings. Given the potential for AI-generated text to be used for malicious purposes, it would be beneficial to have a more in-depth discussion of the ethical implications of the research.

Overall, these weaknesses do not detract from the significant contributions of the paper, but addressing them could further improve the work and its impact.

**Questions:**

Can the authors provide more details on the SpaceInfi strategy, such as how it was developed and why it is effective? This would help readers better understand the strategy and its potential applications.

The paper shows that SpaceInfi is effective in evading detection, but it does not analyze the impact of SpaceInfi on downstream tasks such as question answering or summarization. Can the authors provide more insights into the impact of SpaceInfi on these tasks?

The paper only compares the SpaceInfi strategy with style transfer, but there are other evasion strategies that could be compared to SpaceInfi. Can the authors provide a comparison with other evasion strategies, such as adversarial attacks or other methods that introduce subtle changes to the text?

The paper briefly mentions the potential misuse of AI-generated text but does not delve into the ethical implications of the findings. Can the authors provide a more in-depth discussion of the ethical implications of the research?

The experiments in this paper are limited to a few benchmarks and detectors. Can the authors expand the scope of the experiments to include more benchmarks and detectors to ensure the generalizability of the findings?

The paper shows that general style transfer is ineffective in evading detectors, only when the new style is highly intense can detection potentially be evaded. Can the authors provide more insights into why this is the case and how the intensity of the style affects detector performance?

The paper provides a theoretical explanation for why SpaceInfi is successful in evading perplexity-based detection and empirically shows that a phenomenon called token mutation causes the evasion for language model-based detectors. Can the authors provide more details on the theoretical explanation and how it relates to the empirical findings?

Can the authors provide more insights into the limitations of the SpaceInfi strategy and potential ways to improve it? This would help readers better understand the potential applications and limitations of the strategy.

The paper proposes a new approach to understanding and constructing more applicable ChatGPT detectors. Can the authors provide more insights into how this approach could be applied in practice and its potential impact on the field of natural language processing?

Can the authors provide more insights into the future directions of this research and potential areas for further investigation? This would help readers better understand the potential impact and implications of the research.

---

### Official Review · Reviewer_cd2Y · 2023-10-29

**Soundness:** 2 fair
**Presentation:** 3 good
**Contribution:** 1 poor
**Rating:** 3
**Confidence:** 4

**Summary:**

This paper proposes a very simple method to bypass AI-generated text detectors, which inserts a single space character to the text. Experimental results show that this naive evasion strategy can evade detectors on various LLMs.

**Strengths:**

The experiments are conducted on multiple popular LLMs, including ChatGPT4, Vicuna and Alpaca.

**Weaknesses:**

1.The method is too simple. It should be published as a short paper or a technical report, rather than a long paper.
2.The AI-generated text detectors themselves are not well developed. Thus, attacking them is not that meaningful. From the results in Figure 2, we can see that the style transfer is also effective for evading  detectors. I believe that other simple method like rephrasing can also bypass detectors.

**Questions:**

1.Instead of the simple method of inserting a single space character, is it possible to design a more generic methodology, e.g., using an optimization policy to select the most suitable character and the inserting position?
2.Are existing AI-generated text detectors weak to simple baseline methods, e.g., rephrasing and synonym substitution?

---

### Official Review · Reviewer_7Vtq · 2023-11-02

**Soundness:** 2 fair
**Presentation:** 2 fair
**Contribution:** 2 fair
**Rating:** 3
**Confidence:** 5

**Summary:**

The paper discusses the concerns about the misuse of AI-generated text and the need to detect whether texts are generated by ChatGPT or by humans. It challenges the assumption that there are distributional gaps between human-generated and AI-generated text and proposes the SpaceInfi strategy to evade detection by exploiting subtle differences, such as an extra space. The effectiveness of this strategy is demonstrated through experiments. The paper also provides a theoretical explanation for the success of SpaceInfi in evading perplexity-based detection and highlights the role of token mutation in language model-based detectors. Overall, the findings offer new insights and challenges for understanding and constructing more applicable ChatGPT detectors.

**Strengths:**

1. This paper is well-written.
2. The topic of this paper is essential. Detecting machine-written text against human-written is indeed important.
3. The experiment results seem good.

**Weaknesses:**

1. The paper format needs to be carefully aligned with ICLR 2024 guidelines. For instance, on page 7, the page number is obscured by the table line.
2. While the experimental results seem promising, the underlying reasons for the attack's effectiveness remain unclear. The attack may succeed simply because the detector was trained on data with spaces at the start/end.
3. Further ablation studies should explore if other characters (e.g. "!", "#") also enable attacks like "SPACE".
4. The "theoretical understanding" section requires a more comprehensive analysis of why the attack works, rather than just defining common terms.
5. Consider moving some tables/figures to the appendix to avoid occupying too much space in the main text.

**Questions:**

Please refer to the weakness part.

**Details Of Ethics Concerns:**

No concerns